# Rational Design and In Vivo Characterization of mRNA-Encoded Broadly Neutralizing Antibody Combinations against HIV-1

**DOI:** 10.3390/antib11040067

**Published:** 2022-10-24

**Authors:** Elisabeth Narayanan, Samantha Falcone, Sayda M. Elbashir, Husain Attarwala, Kimberly Hassett, Michael S. Seaman, Andrea Carfi, Sunny Himansu

**Affiliations:** 1Moderna, Inc., Cambridge, MA 02139, USA; 2Center for Virology and Vaccine Research, Beth Israel Deaconess Medical Center, Boston, MA 02215, USA

**Keywords:** HIV, mRNA, antibodies, broadly neutralizing antibodies

## Abstract

Monoclonal antibodies have been used successfully as recombinant protein therapy; however, for HIV, multiple broadly neutralizing antibodies may be necessary. We used the mRNA-LNP platform for in vivo co-expression of 3 broadly neutralizing antibodies, PGDM1400, PGT121, and N6, directed against the HIV-1 envelope protein. mRNA-encoded HIV-1 antibodies were engineered as single-chain Fc (scFv-Fc) to overcome heavy- and light-chain mismatch. In vitro neutralization breadth and potency of the constructs were compared to their parental IgG form. We assessed the ability of these scFv-Fcs to be expressed individually and in combination in vivo, and neutralization and pharmacokinetics were compared to the corresponding full-length IgGs. Single-chain PGDM1400 and PGT121 exhibited neutralization potency comparable to parental IgG, achieving peak systemic concentrations ≥ 30.81 μg/mL in mice; full-length N6 IgG achieved a peak concentration of 974 μg/mL, but did not tolerate single-chain conversion. The mRNA combination encoding full-length N6 IgG and single-chain PGDM1400 and PGT121 was efficiently expressed in mice, achieving high systemic concentration and desired neutralization potency. Analysis of mice sera demonstrated each antibody contributed towards neutralization of multiple HIV-1 pseudoviruses. Together, these data show that the mRNA-LNP platform provides a promising approach for antibody-based HIV treatment and is well-suited for development of combination therapeutics.

## 1. Introduction

The delivery of monoclonal antibodies (mAbs) as recombinant protein therapy has been successful in the prevention or treatment of a variety of conditions [1,2,3,4,5]. For many viruses, delivery of a single-antigen specific antibody is often sufficient to prevent infection or disease. However, for pathogens such as human immunodeficiency virus (HIV) that have high antigen variability and potential for mutational escape, delivery of multiple antibodies may be necessary [6,7,8,9,10]. The safety and efficacy of recombinant monoclonal antibody combinations against HIV-1, and more recently, severe acute respiratory syndrome coronavirus 2, have been evaluated in clinical trials [11,12]. However, the high costs associated with their manufacturing, coformulation, and clinical development often present challenges [13].

The mRNA-lipid nanoparticle (LNP) platform provides a potential solution to the manufacturing and delivery of multiple antibodies as a single drug product [14]. Indeed, the mRNA-LNP platform is ideally suited for in vivo co-expression of multiple proteins [15], and mRNA vaccines that rely on the in vivo expression of multiple proteins are being studied in humans [16], including a cytomegalovirus mRNA vaccine that has completed phase 2 studies [17] and a human metapneumovirus and human parainfluenza virus type 3 combination vaccine under phase 1b evaluation [18]. In addition, an mRNA-based therapeutic encoding for OX40L, IL-23, and IL-36γ is under clinical investigation for the treatment of solid tumors [19]. Recently, a first-in-human study also demonstrated that mRNA delivery of a monoclonal antibody for protection against chikungunya disease was well-tolerated, achieving functionally relevant plasma antibody levels [20].

Here, we present an mRNA-LNP platform as a potential solution to the manufacturing and delivery of multiple antibodies as a single drug product. As a proof of concept of coformulation and delivery of multiple mRNA antibodies in a single drug product, we designed and evaluated a cocktail of 3 broadly neutralizing antibodies (bnAbs) against the envelope (Env) protein of HIV-1. The delivery of multiple HIV-1 bnAbs in an mRNA-LNP platform may lead to simultaneous co-expression of multiple bnAbs within a single cell. This presents the risk of heavy- and light-chain mismatch pairing and assembly of non-functional antibody heterodimers. To overcome this, we engineered several HIV-1 bnAbs as single-chain crystallizable fragment (Fc) variants (scFv-Fcs). A panel of HIV-1 bnAbs that target the V1/V2 loop, the V3 glycan base, and the CD4 receptor binding site (CD4bs) [21,22,23] was selected for construction of the scFv-Fc. We then investigated the ability of each scFv-Fc HIV-1 bnAb to maintain in vitro neutralization breadth and potency compared to their parental immunoglobulin G (IgG) form, as well as their ability to be expressed individually and in combination in vivo. Finally, we determined the ability of the bnAb combination to achieve neutralization and pharmacokinetics (PK) comparable to recombinant full-length IgG.

## 2. Materials and Methods

### 2.1. Recombinant IgG and scFv-Fc Expression, Purification, and Characterization

Recombinant IgG and scFv-Fcs were expressed in Expi293 cells (ThermoFisher, Waltham, MA, USA) and purified using protein A Fast protein liquid chromatography with subsequent preparative size exclusion chromatography (SEC). Proteins were subsequently exchanged into a phosphate-buffered saline (PBS) and characterized using SDS-PAGE, Western blotting, analytical SEC, and the SYPRO orange ThermoFleur assay.

Proteins were characterized as a function of temperature using the ThermoFleur assay through the addition of 5-μM protein to SYPRO orange. Temperature and fluorescence monitoring were performed using a CFX384 Touch machine (Bio-Rad, Hercules, CA, USA), with temperature increased from 25 °C to 90 °C; samples were incubated for 5 s at each temperature prior to fluorescence measurement [24,25].

### 2.2. mRNA-LNP Production and Formulation

A sequence-optimized mRNA was synthesized by in vitro T7 RNA polymerase (New England Biolabs, Ipswich, MA, USA) mediated transcription with complete replacement of uridine by N1-methyl-pseudouridine. The final mRNA encoded the protein of interest, a 5′ untranslated region (UTR), a 3′ UTR, and a DNA-template encoded polyA tail. After transcription, the Cap 1 structure was added to the 5′ end using Vaccinia Capping Enzyme (New England Biolabs) and Vaccinia 2′ O-methyltransferase (New England Biolabs). Lipid nanoparticles were produced via nanoprecipitation by mixing lipids dissolved in ethanol (ionizable lipid, phospholipid, sterol, and polyethylene glycol lipid) with mRNA diluted in 25 mM sodium acetate (pH 5) (Sigma Aldrich, St. Louis, MO, USA) at a ratio of 3:1 (aqueous: ethanol). Formulations were then dialyzed against PBS (pH 7.4) (Lonza, Basel, Switzerland) in a Slide-A-Lyzer dialysis cassette (ThermoFisher, MA, USA) for ≥18 h at 4 °C. Formulations were concentrated using Amicon ultra centrifugal filters (EMD Millipore, Burlington, MA, USA), passed through a 0.22-μm filter (EMD Millipore, MA, USA), and stored at 4 °C. All formulations were tested for particle size, RNA encapsulation, and endotoxins. Formulations were between 80 nm–100 nm, with >90% encapsulation and <10 EU/mL endotoxin and were deemed acceptable for in vivo study.

### 2.3. Animal Experiments

Pharmacokinetic studies were conducted in Tg32 hemizygous mice [neonatal Fc receptor (FcRn)−/− human neonatal Fc receptor (hFcRn) (32) T, FcRn−/− hFcRn (32) Tg, hFcRn Tg32, Tg32; Jackson Laboratories, Bar Harbor, ME, USA]. Recombinant IgGs (2 mg/kg) and mRNA-LNPs (0.3 mg/kg) were administered intravenously via a 50-μL tail-vein bolus injection. Serum samples were collected using cheek bleeds 1, 2, 5, and 18 days following mRNA administration and were tested for total human IgG concentrations using a total human IgG Meso Scale Discovery (MSD) assay. Concentrations of individual HIV bnAbs at each time point were inferred using a pseudovirus neutralization assay, in which viruses sensitive to a single antibody in the administered combination (but resistant to neutralization by the others) were used, as previously described [8]. Serum titers of the median infectious dose (ID_50_) were multiplied by the half-maximal inhibitory concentration (IC_50_) titer of the control antibody against the relevant virus to calculate the estimated serum concentration of each antibody. Animals used were housed and handled ethically as per institutional animal care and use committee (IACUC) protocols, and studies were approved by the Moderna IACUC.

### 2.4. Quantification of IgG in Animal Sera

IgG concentrations in animal sera were measured using a total human IgG MSD assay optimized for detection of full-length human IgGs and scFv-Fcs. Briefly, 96-well MSD plates (Meso Scale Diagnostics, Rockville, MD, USA) were coated overnight with 100 ng/well of goat anti-human variable light-chain (V + L) fragment antigen-binding region (Fab) (Jackson ImmunoResearch, West Grove, PA, USA). Plates were washed and blocked with 5% BLOTTO Blocking buffer (ThermoFisher). Samples were serially diluted in PBS and incubated for 1 h at room temperature, followed by a wash step and the addition of 25 ng/well goat anti-human-FC Sulfo-Tag capture antibody (Meso Scale Diagnostics). Plates were incubated again for 1 h at room temperature, after which another wash step was performed, and plates were developed using MSD read buffer T (Meso Scale Diagnostics). IgG concentrations were determined using a protein reference standard (ThermoFisher).

### 2.5. TZM-bl In Vitro Neutralization Assay

This assay measures antibody-mediated neutralization of the virus as a function of reductions in HIV-1 Tat-regulated firefly luciferase reporter gene expression [26]. Mouse serum samples or purified scFv-Fcs and IgGs were tested in TZM-bl neutralization assays as previously described [26,27]. Briefly, serum samples were tested using a primary 1:50 dilution and secondary 3-fold serial dilution for a total of 8 sample dilutions tested. Purified antibodies were tested using a primary concentration of up to 25 μg/mL. HIV-1 Env pseudovirus was added to wells and incubated (1 h; 37°C) before addition of TZM-bl target cells (1 × 10^4^ cells/well) and DEAE-dextran (11 μg/mL) (Sigma Aldrich). Plates were incubated for 48 h at 37 °C and then harvested and developed using BrightGlo luciferase (Promega, Madison, WI, USA) and a GloMax Navigator luminometer (Promega). Data are reported as the serum dilution or antibody concentration that inhibited 50% or 80% of viral infection (ID_50_ and ID_80_ titers or IC_50_ and IC_80_ titers, respectively). All serum samples were heat-inactivated at 56 °C for 30 min prior to use. Murine leukemia virus pseudovirus was used as a negative control. Purified IgG bnAbs were utilized as positive neutralization controls and were tested as above (10 μg/mL with serial 3-fold dilutions).

### 2.6. Pharmacometric Analysis

Mean serum concentration-time profiles of N6 IgG, PGT121 scFv-Fc Var7B, and PGDM1400 scFv-Fc Var7, quantified following an intravenous dose of coformulated mRNA at 1, 2, and 3 mg/kg at time points up to 18 days after dosing, were used for development of the model. A kinetic-pharmacodynamic (KPD) model was built to describe and predict antibody dynamics. This model included 2 compartments, namely the effect compartment and the central antibody compartment. The effect compartment was expressed as:$$\frac{dAe}{dt}=-Ae\times Ke$$
where *Ae* is the amount of mRNA in the effect compartment and *Ke* is elimination rate of mRNA from the effect compartment. The antibody production rate (*Ksyn*) was modeled as a linear function of the mRNA amount in the effect compartment (*Ae*) as follows:Ksyn=m×Ae
where *Ksyn* is the antibody production rate modeled as the product of linear slope parameter *m* and *Ae*. The in vivo dynamics of antibodies were modeled using a 1-compartment model, expressed as follows:$$\frac{dAb}{dt}=Ksyn-\left(Cab\times CLab\right)$$
where *Ab* and *Cab* are the amount and concentration, respectively, of the antibody in the central compartment (circulation). *CLab* is the antibody clearance rate from the central compartment. PK/pharmacodynamic (PD) model parameters were estimated using a nonlinear mixed-effects modeling approach using Phoenix NLME software, Version 8.3.3 (Certara, Princeton, NJ, USA). The First Order Conditional Estimation-Extended Least Squares (FOCE-ELS) method was employed for all model runs. Datasets and graphics were prepared using R. Assessment of model adequacy was guided by goodness-of-fit criteria, including visual inspection of diagnostic scatter plots (observed vs. predicted concentration, residual/weighted residual vs. predicted concentration or time), successful convergence of the minimization routine, and plausibility of parameter estimates.

The mouse KPD model was scaled allometrically to humans via body weight–based scaling of model parameters. Volume, mRNA clearance, antibody clearance, and protein synthesis rate were scaled using body weight–based allometric exponents of 1, 0.75, 0.93, and 0.8, respectively.

## 3. Results

### 3.1. Design of mRNA-Encoded Antibody Combinations Using Single-Chain Conversion

Antibodies are heterodimeric proteins requiring co-expression of the heavy and light chains, which are encoded on 2 separate open reading frames (Figure 1A, top). However, if mRNA encoding heavy and light chains of 2 or more antibodies are co-delivered to the same cell, the resulting expressed heavy and light chains may misassemble (“scrambling”), resulting in mispaired, non-functional species (Figure 1A, bottom).

To minimize the risk of scrambling, we used a scFv, in which the heavy- and light-chain variable domains of each antibody are physically tethered to each other with flexible linkers. Rosetta modeling software was used for single-chain computational modeling and thermostability predictions. Mutations were introduced and evaluated in each design using the Rosetta scoring function, which is dominated by attractive and repulsive Lennard-Jones interactions, an orientation-dependent hydrogen bonding term [28], and an implicit solvation model [29]. To increase the size and improve half-life, the scFv constructs were further linked to a constant Fc region, resulting in scFv-Fc constructs (Figure 1B). Notably, mRNA-based expression of antibodies in the scFv-Fc format not only eliminates mispairing, resulting in only mono- or bispecific antibodies (Figure 1B, bottom), but also halves the number of mRNAs in the formulation [30]. Finally, to improve the in vivo half-life, the Leucine Serine (LS) mutation (Met424Leu; Asn434Ser) was introduced into the Fc region [31].

The initial scFv-Fc antibody designs (scFv-Fc.r1) focused on the conversion of a single CD4bs antibody (3BNC117), a single V1/V2 bnAb (PGDM1400), and 2 bnAbs directed against the V3-base (PGT121; 10-1074) [21,22,23]. The variable heavy (V_H_) and variable light (V_L_) domains were linked using a 15-amino acid (G_4_S)_3_ linker, and a series of variants were engineered with mutations, with the goal of improving protein stability and expression, as well as minimizing immunogenicity.

### 3.2. Expression of scFv-Fc Antibodies

The tolerance of conversion from IgG format to scFv-Fc can vary from one antibody to another. To maximize chances of success, HIV-1 bnAbs (3BNC117, PGDM1400, PGT121, and 10-1074) were tested for tolerance to conversion to the scFv-Fc format (Figure 2A). Each scFv-Fc design was expressed as a recombinant protein and tested for in vitro expression and potency across a global panel of HIV-1 pseudoviruses [32] selected based on sensitivity to the parental IgG bnAb using the TZM-bl assay. Each scFv design was compared to its parental IgG antibody as a control.

The potency and expression of the lead scFv-Fc.r1 designs and their parental IgG antibodies are shown in Figure 2A. Expression levels of scFv-Fc antibodies relative to their parental antibodies varied (Figure 2A–D). The scFv-Fc.r1 variant based on the V1/V2 bnAb PGDM1400 exhibited an in vitro neutralization profile comparable to that of its parental IgG (Figure 2A,B; Appendix A). However, the scFv-Fc.r1 exhibited a ~1.6-fold reduction in expression levels relative to its parental IgG. In contrast, scFv-Fc.r1 conversion was less successful for the 2 V3-base antibodies. The 10-1074 showed reduced in vitro expression while retaining neutralization potency, whereas PGT121 retained similar expression levels and neutralization potency (Figure 2A,C; Appendix A). Finally, the CD4bs bnAb, 3BNC117, was resistant to scFv-Fc.r1 conversion, with a ~10-fold reduction in potency and a 4-fold reduction in expression compared to its parental IgG (Figure 2A,D; Appendix A).

Based on the initial results of scFv-Fc.r1 designs, a series of new scFv-Fc variants was engineered (scFv-Fc.r2). We hypothesized that the observed reduced expression and potency of some scFv-Fc antibodies could be due to insufficient length of the 15-amino acid VH-VL linker, allowing the interface to open and multimerize into nonfunctional oligomeric diabodies [33]. Therefore, the new scFv-Fc.r2 designs incorporated a longer VH-VL linker (20-amino acid GS linker [G_4_S]_4_). A small set of scFv-Fc variants containing point mutations predicted to increase VH or VL stability and single-chain designs for other CD4bs HIV bnAbs (N6 and NIH 45-46) was also tested.

Upon introduction of the 20-amino acid GS linker, the in vitro expression levels of PGDM1400 and PGT121 single chains was comparable to the 15-amino acid GS linker; however, the resulting proteins had improved monomeric profiles, as observed by SEC analysis (Figure 2B,C; Appendix A). In vitro neutralization of single chains with 20-amino acid linkers was also comparable to the full-length IgG forms (Figure 2B,C; Appendix A). Similar single-chain designs for 3BNC117 showed improved in vitro expression, but still had lower in vitro neutralization potency and/or breadth compared to its parental IgG form (Figure 2A,D; Appendix A). Designs for 10-1074 still showed a ~6-fold reduction in expression levels compared to parental IgG (Figure 2A,C) and were no longer pursued due to the success with the PGT121 single-chain designs (Figure 2C). For PGDM1400, although there were several round 2 scFv-Fc point mutations with favorable expression and potency, we selected the wild-type VH-VL, with a 20-amino acid linker as the top candidate (scFv-Fc.r2; Figure 2A), as expression of PGDM1400 increased from 200 μg/mL for the scFv-Fc.r1 variant to 340 μg/mL for the scFv-Fc.r2 variant, with potency remaining comparable to the parental IgG (Figure 2B; Appendix A). Size exclusion chromatography analysis revealed that most of the eluted protein was in a single peak (Figure 2B), indicating little or no aggregation or multimerization.

Since the single-chain conversion for our CD4bs bnAb 3BNC117 performed poorly in the first round, we considered 2 additional CD4bs antibodies in the second round: the highly potent N6 and NIH45-46 antibodies. Although expression was sufficient for 2 of the 3 round 2 CD4bs scFv-Fc designs, the in vitro neutralization potency was compromised for all 3 single-chain CD4bs bnAbs by 4- to 7-fold (Figure 2A,D; Appendix A). Since all 3 CD4bs antibodies failed to perform well in the single-chain format, the full-length IgG was chosen for the CD4bs component of the bnAb combination. N6 IgG was selected due to its remarkable neutralization breadth and potency [23]. Round 2 designs containing additional mutations for all antibodies (Appendix A) were engineered to improve protein thermostability and obtain in vitro expression levels comparable to their unmutated scFv-Fc counterparts. However, slight differences in in vitro expression were observed among point mutation variants, as well as small improvements in the neutralization profiles (Figure 2; Appendix A).

### 3.3. In Vivo Expression of Individual Components of Candidate HIV mRNA-LNP bnAbs

We previously demonstrated that mRNAs encoding IgG subunits delivered via mRNA-LNPs lead to production of full-length IgG in mice [34] and in humans [20]. Prior to testing a bnAb mRNA-LNP cocktail in vivo, we assessed protein expression levels of each individual bnAb in mice using a total human IgG enzyme-linked immunosorbent assay. All scFv-Fc designs were found to be expressed at elevated levels, with the leading round 2 designs for PGT121 and PGDM1400 achieving systemic concentrations of ≥73.33 μg/mL at day 1 and ≥30.81 μg/mL at day 5, after a 10-μg (0.5 mg/kg) intravenous dose of mRNA-LNP (Figure 3). Full-length N6 IgG (0.5 mg/kg) achieved peak antibody titers (day 1 = 974 μg/mL; day 5 = 260 μg/mL) several fold higher than those of the leading PGT121 and PGDM1400 scFv single-chain designs (Figure 3). Data are not shown for other variants.

### 3.4. Selection of a Lead HIV bnAb mRNA-LNP Combination

The following combination was selected for further testing in vivo: N6 full-length human IgG, PGDM1400 scFv-Fc.r2, and PGT121 scFv-Fc.r2. The lead candidates were selected based on recombinant protein expression levels, monomeric profiles in SEC, neutralization breadth and potency against relevant HIV pseudoviruses, and in vivo expression of antibodies in Balb/c mice (Figure 2 and Figure 3). Lead mRNA-LNPs were designed to contain either a human heavy-chain signal sequence (N6 heavy chain) or a human kappa light-chain signal sequence (N6 light chain or scFv-Fcs). In addition, N6 full-length IgG and single-chain designs for PGDM1400 and PGT121 contained human IgG1 Fc with LS half-life extension mutations.

### 3.5. In Vivo Expression of an mRNA-LNP Encoded HIV bnAb Combination

We next sought to understand the PK/PD of the mRNA-LNP-encoded HIV bnAb combination and determine whether therapeutically relevant expression levels can be achieved. Four mRNAs encoding PGT121 scFv-Fc.r2, PGDM1400 scFv-Fc.r2, and N6 IgG heavy chain and N6 light chain (heavy:light, 2:1) were coformulated in LNPs at a 1:1:1:0.5 ratio. The mRNA-LNP combination was tested for in vivo expression in Tg32 hemizygous mice at multiple doses. IgG1 PK in Tg32 hemizygous mice have been shown to better predict human IgG1 PK compared to mice expressing mouse FcRn [35]. Individual components of the HIV mRNA bnAb cocktail and recombinant IgG forms of each bnAb were also tested as controls; data from mouse sera indicate that the levels of protein expression achieved from the mRNA bnAb were comparable to the recombinant IgG (Figure 4A–C). Plasma protein expression levels of each individual bnAb in mice were extrapolated using an in vitro neutralization assay. To extrapolate protein expression levels, reference standards of purified IgG were added to mouse serum and tested alongside each bnAb. Differences in plasma protein expression levels compared to those obtained when the bnAbs were administered separately may be due to differences in the assays used to measure protein concentration.

To assess half-life in plasma, antibody concentration versus time data were adequately fitted to the KPD model (Appendix A). The modeled human half-life of N6 IgG mRNA was 2 days longer than the full-length IgG; the half-life of PGT121 scFv-Fc and PGDM1400 scFv-Fc was 6 and 23 days longer than the recombinant full-length IgG, respectively (Figure 4D). In addition, model-based extrapolation of antibody concentrations suggested that dosing of the mRNA combination at 1 mg/kg (0.3 mg/kg per bnAb) could maintain protein levels of ≥10 μg/mL for each antibody up to 90 days following infusion in mice (Figure 5). Peak concentrations after mRNA administration of the combination at 0.3 mg/kg of each bnAb (Figure 5) is approximately one third of those observed at 1 mg/kg (Figure 4). Based on these data, we predict that when combined, these antibodies will be expressed at similar levels and maintain their individual potency since the concentration of each antibody within the combination has previously been determined through extrapolation of neutralization data from antibody sensitive virus strains. The KPD model was scaled allometrically to humans to predict antibody concentrations following an IV dose of mRNA combinations. The mRNA combination dose needed to maintain ≥10 μg/mL concentration for at least 90 days after infusion in humans was predicted to be 0.2 mg/kg.

## 4. Discussion

Our data demonstrate that the mRNA-LNP platform can be used to express in vivo multiple complex proteins, including intact IgG and single-chain antibody combinations, from a single formulation. Delivery of multiple IgGs using mRNA-LNP technologies is best suited for single-domain antibodies or antibodies with matched light chains, as single-chain conversion of full-length IgGs can lead to reduced potency. For HIV-1 bnAbs, 3BNC117, N6, and NIH45-46 lost in vitro neutralization potency and/or breadth after single-chain conversion, potentially due to the impact of single-chain conversion on protein structure, leading to reduced binding of HIV-1 envelope proteins. However, single-chain versions of PGDM1400 and PGT121 exhibited potency and biophysical properties comparable to parental IgG forms.

The use of bnAbs offers a promising approach to HIV treatment and has been shown to be highly effective in preventing HIV-1 transmission and progression. To date, several bnAbs have entered clinical trials, including VRC01 [36], 3BNC117 [37], 10-1074 [38], PGT121 [9], N6LS, PGDM1400 [39], and 10E8.4/iMab. However, there are several challenges to their use, including transient suppression of viremia, emergence of resistance, and reduced efficacy in cell-to-cell viral transmission [40]. In addition, pathogens having high antigen variability and potential for mutational escape may require the delivery of multiple antibodies. To the best of our knowledge, this is the first such study in which an mRNA-LNP platform has been developed to deliver multiple antibodies in a single drug product.

The 4 mRNA combination encoding the full-length N6 IgG and single-chain versions of PGDM1400 and PGT121 were efficiently expressed in mice and achieved high systemic concentrations. Serum samples demonstrated the contribution of each mRNA-LNP–expressed antibody against a panel of HIV pseudoviruses. Furthermore, studies in transgenic mice confirmed that mRNA-encoded HIV bnAbs have PK profiles similar to their full-length IgG forms.

Modeling data suggest that our combination of HIV bnAbs could maintain levels ≥ 10 μg/mL for up to 90 days after a 1-mg/kg intravenous dose of the mRNA cocktail in mice. At the time of experimentation, prevailing literature suggested that protein expression levels in mice > 10 μg/mL (or 10× the ID_80_ of HIV bnAbs [~1 μg/mL]) would be an adequate hypothetical target to allow for antibody neutralization of HIV-1 [41,42,43]. However, more recently, studies have shown that this target may be substantially higher [44] and the use of more potent antibodies may be needed.

This study also provides evidence that mRNA-LNP platforms is well-suited for combination therapeutics that may be a challenge with traditional recombinant technologies. The mRNA-LNP platform is ideally suited for in vivo co-expression of multiple proteins and subsequent expression of therapeutic levels of antibodies. The engineered mRNA-encoded scFv-Fc variants allow for reduced risk of heavy- and light-chain mismatch pairing and assembly of nonfunctional antibody heterodimers, offering the simultaneous translation and co-expression of multiple bnAbs within a single cell.

Recent clinical trials using HIV bnAbs have demonstrated that antibody-based treatments are effective at preventing HIV infection and may suppress viremia in HIV-positive individuals [44]. The rapid development and approval of safe and effective mRNA-based COVID-19 vaccines, as well as studies showing the mRNA-LNP platform to be safe for the delivery of human monoclonal antibodies [20,45], highlight the ability to adapt this platform to produce pharmaceuticals to combat difficult-to-treat diseases.

## Figures and Tables

**Figure 1 antibodies-11-00067-f001:**
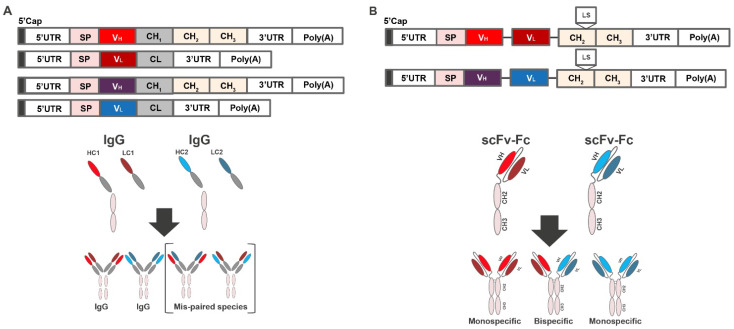
Combinatorial delivery of mRNA encoding 2 antibody products. (**A**) mRNA construct encoding heavy and light chains of 2 or more antibodies and schematic showing production of mispaired species. (**B**) mRNA construct encoding heavy- and light-chain variable domains of each antibody physically tethered with flexible linkers, and the scFv constructs linked to a constant Fc region and schematic showing the production of correctly paired species. CH, heavy chain constant domain; CL, light chain constant domain; Fc, crystallizable fragment; HC, heavy chain; IgG, immunoglobulin G; LC, light chain; Poly(A), polyadenylic acid tail; scFv-Fc, single-chain Fc variant; UTR, untranslated region; VH, variable heavy; VL, variable light.

**Figure 2 antibodies-11-00067-f002:**
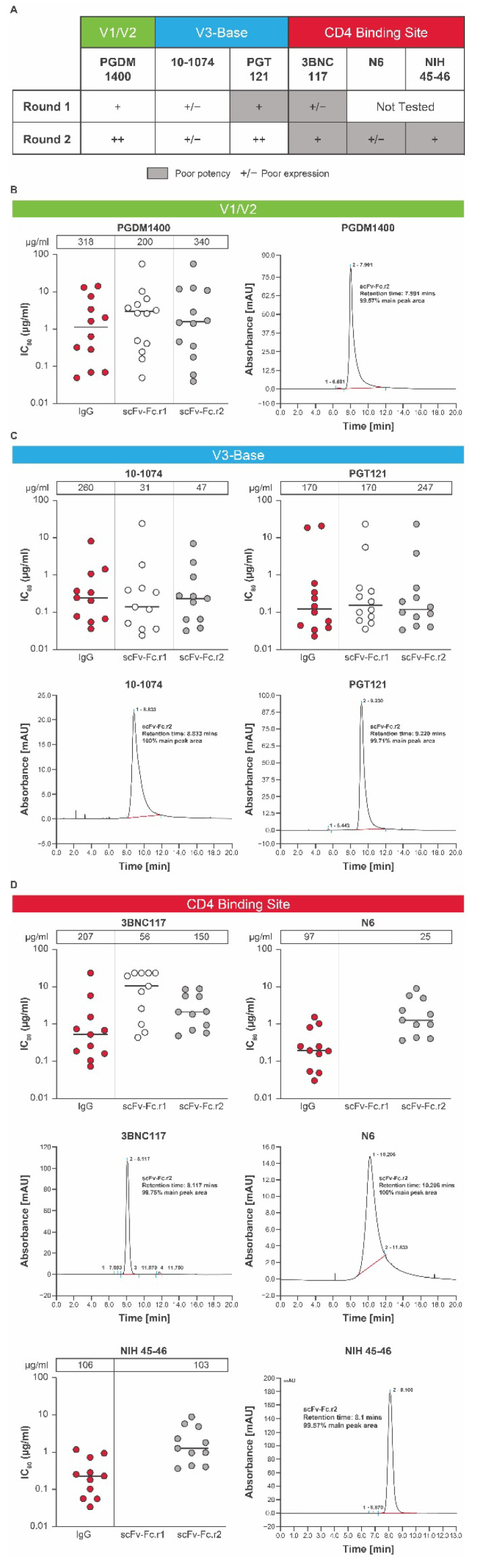
Conversion into scFv-Fcs of individual HIV bnAbs. (**A**) A panel of HIV bnAbs (scFv-Fc.r1 and scFv-Fc.r2 designs) targeting 3 key epitopes was tested for in vitro expression and potency. (**B**) In vitro expression and potency of a PGDM1400 variant with size exclusion chromatography, (**C**) V3-base variants 10-1074 and PGT121 with size exclusion chromatography, and (**D**) CD4bs bnAbs 3BNC117, N6, and NIH45-46 over 2 rounds of modifications (round 1 and round 2). In vitro expression concentrations (μg/mL) of the scFv-Fc.r1 and scFv-Fc.r2 designs and their parental IgG antibodies are shown above each figure. Each antibody was tested using a panel of HIV-1 Env pseudoviruses selected based on neutralization sensitivity of the parental bnAb. Protein expression for each mAb and single chain was completed twice. SEC and neutralization analyses were only performed on large scale samples with testing in duplicates each performed once. Note: the lower the IC_80_, the greater the potency. bnAb, broadly neutralizing antibody; Env, envelope; Fc, crystallizable fragment; HIV, human immunodeficiency virus; IC_80_, inhibitory concentration of 80%; IgG, immunoglobulin G; scFv-Fc, single-chain variable fragment; SEC, size exclusion chromatography.

**Figure 3 antibodies-11-00067-f003:**
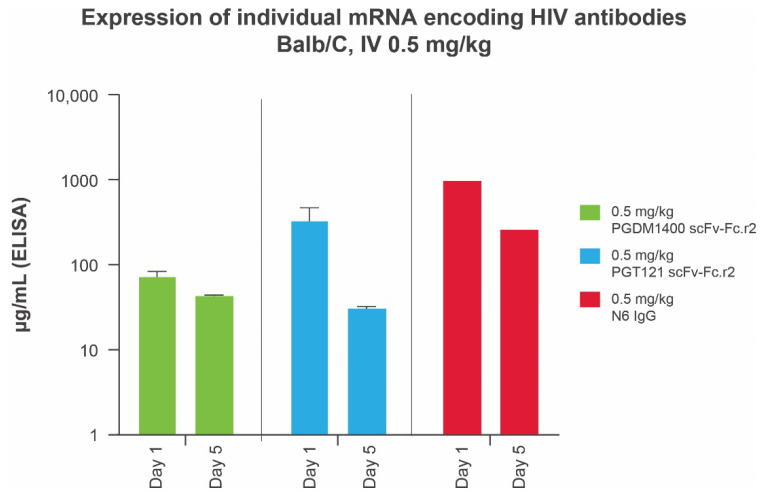
Serum expression of individual mRNA encoding IgG and scFv-Fcs in Balb/C mice (*n* = 10 per group) at days 1 and 5. IgG concentrations in animal sera were measured using a total human IgG MSD assay optimized for detection of full-length human IgGs and single-chain scFv-Fcs. ELISA, enzyme-linked immunosorbent assay; Fc, crystallizable fragment; HIV, human immunodeficiency virus; IgG, immunoglobulin G; IV, intravenous; MSD, Meso Scale Discovery; scFv-Fc, single-chain variable fragment.

**Figure 4 antibodies-11-00067-f004:**
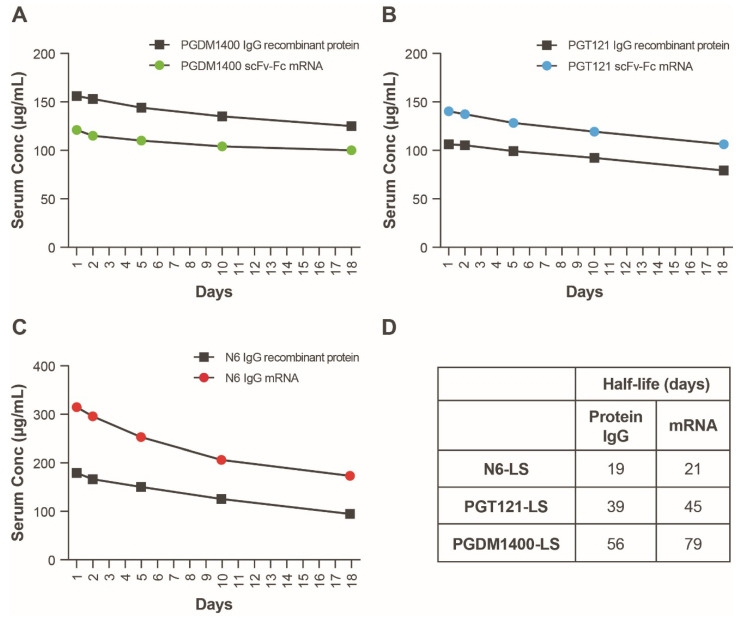
Serum expression of individual mRNAs encoding scFv-Fcs or IgGs as compared to recombinant protein. (**A**–**C**) Antibodies were delivered either as recombinant protein (squares, 10 mg/kg) or mRNA encapsulated LNPs (circles, 1 mg/kg). Serum concentrations of all constructs were back-extrapolated based on serum neutralizing ID_50_ titers using sensitive strains of HIV-1 Env pseudovirus. Each of 3 HIV bnAbs, (**A**) PGDM1400, (**B**) PGT121, and (**C**) N6, were administered separately to 5 FcRN transgenic mice, and serum was pooled and tested for neutralization on days 1, 2, 5, 10, 18, and 24. (**D**) Modeled human half-lives of scFv-Fcs and full-length IgG containing the LS mutation. bnAb, broadly neutralizing antibody; Fc, crystallizable fragment; Env, envelope; HIV, human immunodeficiency virus; ID_50_, median infectious dose; IgG, immunoglobulin G; LNP, lipid nanoparticle; LS, Leucine Serine; scFv, single-chain variable fragment.

**Figure 5 antibodies-11-00067-f005:**
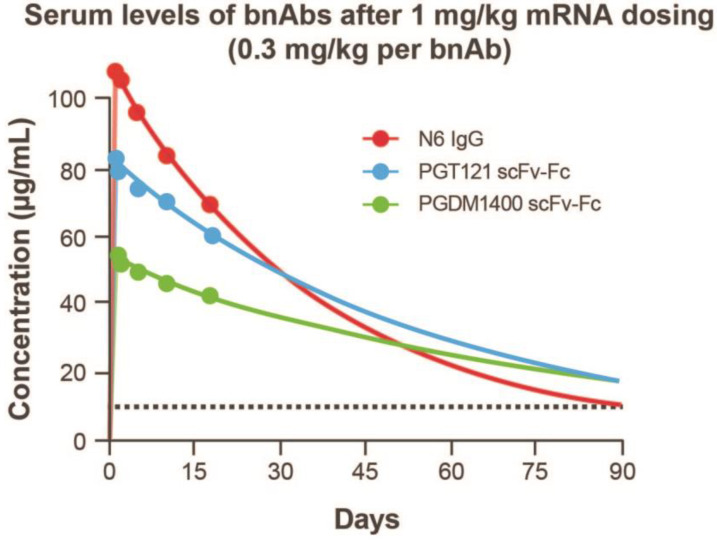
Kinetic-pharmacodynamic model characterization of mRNAs encoding 3 HIV antibodies. Extrapolated serum levels of bnAbs after 1 mg/kg mRNA dosing up to 90 days following infusion. bnAb, broadly neutralizing antibody; Fc, crystallizable fragment; HIV, human immunodeficiency virus; IgG, immunoglobulin G; scFv, single-chain variable fragment.

## Data Availability

Upon request, and subject to review, Moderna, Inc., will provide the data that support the findings of this study.

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
