# Peer review of "Rational Design and In Vivo Characterization of mRNA-Encoded Broadly Neutralizing Antibody Combinations against HIV-1"

_2073-4468, 2022, doi:10.3390/antib11040067_

Round 1

Reviewer 1 Report

In this manuscript, the authors described the first study to utilize the mRNA-LNP platform to co-express multiple HIV bnAbs as a single drug product in vivo. The authors first provided a thorough introduction of the mRNA-LNP application for in vivo expression of multiple proteins for prophylaxis or therapeutic purposes, leading to their own study of applying this platform for expressing three monoclonal antibodies targeting distinct epitopes on HIV Env, by incorporating into the scFv-Fc constructs to tether heavy and light chain to prevent mismatch events in vivo. Neutralization breadth and potency as well as their pharmacokinetics were carefully compared between different versions of antibody constructs. This manuscript is well-written and is of high research and clinical relevance, providing important insights into the development of potential therapeutic platforms. 

The results showed that the conversion of the IgG versions into scFv-Fc will reduce neut potency, I’m wondering whether the authors can add a simple binding experiment to look at the differences? If the binding is strong enough, maybe these scFv-Fc can mediate Fc-receptor functions, for example, ADCC etc., which can still provide protection against virus in vivo? So any possible experiments or discussion about the Fc function of these scFv-Fc may provide an additional aspect of this manuscript.

Mice serum after mRNA injection expressing the combination of the 3 bnAbs were directly tested against a panel of pseudoviruses for extrapolated estimate of the mAb amount circulating in the blood, however, these extrapolated serum bnAb level are not comparable to the levels seen in the experiments when they were administered separately. For example, in figure 4, the N6-IgG showed the highest level of serum expression level, while after administered in combination with the other two bnAbs, N6-IgG became the lowest expressed one. Can the authors discuss about this difference in several sentences?

Authors may use one or two sentences mentioning how long after mRNA injection will the mAbs get expressed and what’s the short-term dynamic.

Here the the Fc from IgG1 was utilized for the constructs, have the authors thought about the Fc from other isotypes, or other IgG subclasses (IgG2 or 3 or 4)?

How did the authors decide the comparable dose of protein vs mRNA for injection?

At the first appearance of the word “HIV”, it may be necessary to include its full name.

Author Response

We would like to thank the Reviewers for their review and suggestions to improve the manuscript. Below are our point-by-point responses, with corresponding edits to the manuscript tracked, as requested. Please note that the line numbers referenced in our responses correspond to those in the tracked manuscript.

Reviewer 2:

In this manuscript, the authors described the first study to utilize the mRNA-LNP platform to co-express multiple HIV bnAbs as a single drug product in vivo. The authors first provided a thorough introduction of the mRNA-LNP application for in vivo expression of multiple proteins for prophylaxis or therapeutic purposes, leading to their own study of applying this platform for expressing three monoclonal antibodies targeting distinct epitopes on HIV Env, by incorporating into the scFv-Fc constructs to tether heavy and light chain to prevent mismatch events in vivo. Neutralization breadth and potency as well as their pharmacokinetics were carefully compared between different versions of antibody constructs. This manuscript is well-written and is of high research and clinical relevance, providing important insights into the development of potential therapeutic platforms.

  • The results showed that the conversion of the IgG versions into scFv-Fc will reduce neut potency, I’m wondering whether the authors can add a simple binding experiment to look at the differences? If the binding is strong enough, maybe these scFv-Fc can mediate Fc-receptor functions, for example, ADCC etc., which can still provide protection against virus in vivo? So any possible experiments or discussion about the Fc function of these scFv-Fc may provide an additional aspect of this manuscript.

Response: Thank you for the comment. Binding data are not currently available. This study focused on conversion of bnAbs to scFv for ease of mRNA coadministration. The criteria to qualify a single chain as a suitable design included a >4-fold reduction in neutralization potency and loss of protein stability (as measure of >1 peak (95%) in SEC analysis). Please note, all designs tested had a significant neutralization breadth and potency, suggesting that the antibodies were still able to bind HIV env with high affinity and neutralize the virus, albeit with slightly lower potency. Since no mutations were made to the Fc regions, we suspect these antibodies would preserve their effector function activity.

  • Mice serum after mRNA injection expressing the combination of the 3 bnAbs were directly tested against a panel of pseudoviruses for extrapolated estimate of the mAb amount circulating in the blood, however, these extrapolated serum bnAb level are not comparable to the levels seen in the experiments when they were administered separately. For example, in figure 4, the N6-IgGshowed the highest level of serum expression level, while after administered in combination with the other two bnAbs, N6-IgGbecame the lowest expressed one. Can the authors discuss about this difference in several sentences?

Response: Thank you for the comment. IgG expression levels in mouse studies where antibodies were tested individually were measured using a total human IgG ELISA; however, when antibodies were co-expressed, IgG concentration for each bnAb was calculated using neutralization assays since idiotypes were unavailable at the time of this study. We recognize differences in the assays measuring protein concentration as well as potential study to study variability as the source of these differences. Additional text has been added to lines 327-330 noting these differences

Revised text: “Differences in plasma protein expression levels compared to those obtained when the bnAbs were administered separately may be due to differences in the assays used to measure protein concentration.”

  • Authors may use one or two sentences mentioning how long after mRNA injection will the mAbs get expressed and what’s the short-term dynamic.

Response: Thank you for the comment. In the manuscript, we discuss the modeled human half-life of N6 IgG mRNA as well as PGT121 scFv-Fc and PGDM1400 scFv-Fc. Additionally, we discuss the use of model-based extrapolation to determine the number of days following infusion that protein levels could be maintained above 10 µg/mL. This discussion can be found on lines 342 to 348 of the revised manuscript.

  • Here the the Fc from IgG1 was utilized for the constructs, have the authors thought about the Fc from other isotypes, or other IgG subclasses (IgG2 or 3 or 4)?

Response: Thank you for the comment. We are currently investigating other subclasses and isotypes. The findings from these studies will form part of a separate independent publication.

  • How did the authors decide the comparable dose of protein vs mRNA for injection?

Response: A dose range finding study in mice was performed for each antibody and mRNA prior to a head-to-head study to identify a dose which would provide approximately the same systemic Cmax.

  • At the first appearance of the word “HIV”, it may be necessary to include its full name.

Response: Thank you for the comment. “Human immunodeficiency virus (HIV)” has been added at first mention of HIV.

Reviewer 2 Report

The authors here make a good case for using mRNA-LNP platform for the delivery of multiple antibodies as a single drug product.

The authors however refrain from citing any work on the mRNA-LNP done by the pioneering groups in Genevant Sciences and Arbutus Biopharma. They even have patent around this technology. Would be nice to mention the inventors.

Some of the initial results need to be reported well (find below the explanations). Authors claim of have run Rosetta modeling and aso claim that they have made mutational variants but do not provide any details to back these claims.

Find below the corrections suggested and explanations requested:

Line 39: "However, the high costs associated with their manufacturing", not manufacture.

Line 190: Have the authors used any other linkers like ST linkers, apart from (G4S)3 and (G4S)4? 

 Line 191: Details about computational modeling and Tm predictions are not mentioned.

 Line 195: What was the fraction of bi-specific Ab product formed? How were they removed?

Line 199: Authors use scFv-Fc.r1 and scFc-Fc.r1 interchangeably. Needs correction. This can also be found in lines 222 and 228.

Line 202/203: What mutations were made? Mutations in CDRs or FRs? Is this info mentioned elsewhere (in any other publication)? If yes, put reference. There is no information mentioned about the engineered variants. If the authors mention them, they need to provide this information as well.

Line 210: What HIV pseudo-viruses were selected? Put reference please (or add information).

Figure 2B: Details about which round scFv-Fc this SEC profile belongs to, is missing. Although mentioned in text (line #260), can the authors mention it as they did in 2C. Also the y-axis is missing

Figure 2D: Concomitant SEC profiles missing from 2D. Any reason for not providing the profile for 3BNC117?

Line 230: Which variant are the authors talking about? More details required.

Line 231: scFv-Fc.r1 variant of PGDM1400 shown in figure 2B shows 37% lower yield (200 mg/L Vs. 318 mg/L) as compared to parental IgG. How do the authors claim that the expression profile is comparable?

Line 234: What is the standard error between IgG and scFv-Fc.r1 of PGT121? I don't think authors can really claim that the potency was reduced.

Line 236: It looks like it is >10-fold reduction in potency and around 4-fold reduction in expression.

Line 239-241: Did the authors observe similar results by Rosetta computational modeling of the scFvs (as mentioned in line #191)? Can the authors comment about it?

Line 242/243: The authors make it sound like this was a sequential endeavor. If this is true then how can the authors compare and claim expression and potency differences between scFv-Fc.r1 and scFv-Fc.r2 variants? Can the authors comment on the reproducibility of the production and potency assays (how many times were these repeated)? 

Line 248: y-axis missing on SEC profiles. Also where is the representative round 1 SEC for comparison with round 2? This claim by authors need to be backed by data

Line 249: 2A does not have any SEC profile

Line 251: Can authors provide in vitro neutralization values in a table?

Line 252-254: Somehow this argument from authors doesn't look convincing. Authors can emphasize more on the >8-fold drop in yield for scFv-Fc.r1 which scFv-Fc.r2 (with >5-fold drop) didn't resurrect and hence rule 10-1074 binder out. Otherwise, scFv-Fc.r1 of 10-1074 has potency even better than the IgG version.

Line 255-256: As compared to the IgG or scFv-Fc.r1? Authors must be comparing it with scFv-Fc.r1, but for reader's benefit, it is better to mention it explicitly here as well.

Line 260It is strongly suggested to add a table mentioning retention time (in min) and % monomer content of the desired peak for each of the IgGs and scFv-Fcs tested (both r1 and r2 variants)

Line 268-270: Restructuring of the sentence required.

Line 271-273Authors should provide the information about what mutations were made (in the final chosen r2 variants) and the criteria of their selection. A table is highly recommended.

Line 276: Correct font size.

Line 280/281: Are the authors concerned about the rate of decrease in serum concentration of PGT121 scFv-Fc.r2 variant at d5 post-injection? Can hepatic accumulation be a reason for this decrease as systemically delivered mRNA–LNP complexes mainly target the liver? Can authors comment on this?

Have the authors tried (or are they going to try)  intradermal/intramuscular/subcutaneous administration of mRNA-LNPs as administration via these routes have shown to produce prolonged protein expression?

Line 282: Was dosing of N6 IgG mRNA-LNP the same (0.5 mg/kg)?

Line 284: How many variants of each were taken into in vivo expression?

Line 292-295: Have the authors measured the self-aggregation propensity of the chosen bnAbs (for example by AC-SINS)?

Line 307: Just out of curiocity, why did the authors use hemizygous Tg32 mice instead of homozygous Tg32 (hFcRn)?

Overall, the data presented by the authors adds value to the mRNA-LNP landscape and successfully shows the in vivo delivery of mRNA-LNPs and production of multiple antibodies (IgGs and scFv-Fcs). The anti-HIV1 antibodies targeting different epitopes of HIV1 successfully neutralized the pseudovirus in vitro. The authors however did not test the protection offered by such passive immunization by challenging (mRNA-LNP)-injected mice with HIV virus (to demonstrate the in vivo efficacy offered by such injections). That would have been really convincing

Author Response

We would like to thank the Reviewers for their review and suggestions to improve the manuscript. Below are our point-by-point responses, with corresponding edits to the manuscript tracked, as requested. Please note that the line numbers referenced in our responses correspond to those in the tracked manuscript.

Reviewer 1:

The authors here make a good case for using mRNA-LNP platform for the delivery of multiple antibodies as a single drug product. The authors however refrain from citing any work on the mRNA-LNP done by the pioneering groups in Genevant Sciences and Arbutus Biopharma. They even have patent around this technology. Would be nice to mention the inventors. Some of the initial results need to be reported well (find below the explanations). Authors claim of have run Rosetta modeling and aso claim that they have made mutational variants but do not provide any details to back these claims. Find below the corrections suggested and explanations requested:

Response: We thank the reviewer for their insightful feedback on the manuscript. Based on this feedback, we have revised the manuscript accordingly and have responded to the comments in a point-by-point fashion below. As suggested, we have referenced a study conducted by Thran et al (2017) that discusses expression of mRNA-encoded antibodies in-vivo. This study is shown as reference 15 in the revised manuscript and has been cited on line 45.

  • Line 39: "However, the high costs associated with their manufacturing", not manufacture.

Response: The text has been updated as suggested

  • Line 190: Have the authors used any other linkers like ST linkers, apart from (G4S)3 and (G4S)4?

Response: Thank you for this comment. Other linkers were not tested in this study.

  • Line 191: Details about computational modeling and Tm predictions are not mentioned.

Response: Thank you for this comment. The computational modeling has been discussed in the methods section titled “Pharmacometric analysis” (lines 145-17 of the revised manuscript). We have also included additional text and references in the revised manuscript providing details on the Rosetta energy function (lines 194-197). Additionally, we have noted that the round 2 designs contained additional mutations for all antibodies to improve protein thermostability and improve protein expression levels (lines 280-283). A table showing these mutations has also been included in the supplement (Table S2).

Revised text: “Mutations were introduced and evaluated in each design using the Rosetta scoring function, which is dominated by attractive and repulsive Lennard-Jones interactions, an orientation-dependent hydrogen bonding term28, and an implicit solvation model29.”

Revised text:Round 2 designs containing additional mutations for all antibodies (Table S2) were engineered to improve protein thermostability and obtain in vitro expression levels comparable to their unmutated scFv-Fc counterparts”

  • Line 195: What was the fraction of bi-specific Ab product formed? How were they removed?

Response: We have not characterized or measured the production of bispecific antibodies. However, we hypothesize that bi-specifics may form when multiple single chains are co-expressed using mRNA.

  • Line 199: Authors use scFv-Fc.r1 and scFc-Fc.r1 interchangeably. Needs correction. This can also be found in lines 222 and 228.

Response: The text has been corrected throughout to read “scFv”

  • Line 202/203: What mutations were made? Mutations in CDRs or FRs? Is this info mentioned elsewhere (in any other publication)? If yes, put reference. There is no information mentioned about the engineered variants. If the authors mention them, they need to provide this information as well.

Response: Thank you for this comment. We tried several mutations in the framework region for each protein to increase thermostability and minimize putative human T cell epitopes using Rosetta and Epibase platforms. Several mutants were designed for each antibody; however, the mutations have not been described in the manuscript as they did not significantly impact protein expression and potency and were thus deemed unviable. A supplementary table (Table S2) has been included showing the mutations for the second round of variants.

  • Line 210: What HIV pseudo-viruses were selected? Put reference please (or add information).

Response: The scFv’s were tested against a panel of pseudoviruses selected from the a global panel as previously described in decamp A et al. J Virol. 2014 Mar;88(5):2489-507. We have revised the text to note that this was a global panel (line 215) and have included a reference for the panel of pseudoviruses (reference 32).

Revised text: “Each scFv-Fc design was expressed as a recombinant protein and tested for in vitro expression and potency across a global panel of HIV-1 pseudoviruses30”

  • Figure 2B: Details about which round scFv-Fc this SEC profile belongs to, is missing. Although mentioned in text (line #260), can the authors mention it as they did in 2C. Also, the y-axis is missing

Response: The figure has been updated to indicate that the SEC profile belongs to the second round of variants (scFv-Fc.r2).

  • Figure 2D: Concomitant SEC profiles missing from 2D. Any reason for not providing the profile for 3BNC117?

Response: All SEC profiles have now been included for Figure 2.

  • Line 230: Which variant are the authors talking about? More details required.

Response: Thank you for the comment. This sentence has been reworded for clarity.

Revised text: The scFv-Fc.r1 variant based on the V1/V2 bnAb PGDM1400 exhibited an in vitro neutralization profile comparable to those of its parental IgG (Fig. 2A, B)”

  • Line 231: scFv-Fc.r1 variant of PGDM1400 shown in figure 2Bshows 37% lower yield (200 mg/L Vs. 318 mg/L) as compared to parental IgG. How do the authors claim that the expression profile is comparable?

Response: Thank you for the comment. We have updated the sentence to note that the neutralization profiles were comparable and have expanded on the difference in expression levels between the scFv-Fc.r1 variant and the parental IgG.

Revised text: The scFv-Fc.r1 variant based on the V1/V2 bnAb PGDM1400 exhibited an in vitro neutralization profile comparable to those of its parental IgG (Fig. 2A, B); however, the scFv-Fc.r1 exhibited a ~1.6-fold reduction in expression levels relative to its parental IgG”

  • Line 234: What is the standard error between IgG and scFv-Fc.r1of PGT121? I don't think authors can really claim that the potency was reduced.

Response: We have revised this sentence to remove this claim.

Revised text:The 10-1074 showed reduced in vitro expression while retaining neutralization potency, whereas PGT121 retained similar expression levels and neutralization potency (Fig. 2A, C).”

  • Line 236: It looks like it is >10-fold reduction in potency and around 4-fold reduction in expression.

Response: Thank you for the comment. We have revised this sentence for clarity.

Revised text:Finally, the CD4bs bnAb, 3BNC117, was resistant to scFv-Fc.r1 conversion, with a ~10-fold reduction in potency and a ~4-fold reduction in expression compared to its parental IgG (Fig. 2A, D).”

  • Line 239-241: Did the authors observe similar results by Rosetta computational modeling of the scFvs (as mentioned in line #191)? Can the authors comment about it?

Response: All designs were generated using modeling for favorable delta G calculations.

  • Line 242/243: The authors make it sound like this was a sequential endeavor. If this is true then how can the authors compare and claim expression and potency differences between scFv-Fc.r1 andscFv-Fc.r2 variants? Can the authors comment on the reproducibility of the production and potency assays (how many times were these repeated)?

Response: Thank you for the comment. Based on the initial results obtained for the scFv-Fc.r1 variants, the scFv-Fc.r2 variants were engineered using a longer VH-VL linker in an attempt to improve expression and potency. The expression and potency obtained for these new scFv-Fc.r variants were then compared to that obtained for the round one variants to determine if the longer VH-VL linker had any effect on potency and expression levels. Protein expression for each mAb and single chain was completed twice (once on a small scale and once on a larger scale). SEC and neutralization analyses were only performed on large scale samples with testing in duplicates, each performed once due to large volume of variants and pseudovirus panels. This has now been noted in the Figure 2 legend on lines 229-231.

Revised text: “Protein expression for each mAb and single chain was completed twice. SEC and neutralization analyses were only performed on large scale samples with testing in duplicates each performed once.”

  • Line 248: y-axis missing on SEC profiles. Also where is the representative round 1 SEC for comparison with round 2? This claim by authors need to be backed by data.

Response: SEC profiles are only available for round 2 designs. The Y Axes of the SEC profiles shown in Figure 2 have now been updated.

  • Line 249: 2A does not have any SEC profile

Response: The text has been updated to replace “2A” with “2B” in the sentence.

Revised text: “…however, the resulting proteins had improved monomeric profiles, as observed by SEC analysis (Fig.2B, C)”

  • Line 251: Can authors provide in vitro neutralization values in a table?

Response: A table showing the mean (SD) and median (IQR) in vitro neutralization values has been included in the supplementary information (Table S1). This table has been referred to in lines 239-286 of the revised manuscript.

  • Line 252-254: Somehow this argument from authors doesn't look convincing. Authors can emphasize more on the >8-fold drop in yield for scFv-Fc.r1 which scFv-Fc.r2 (with >5-fold drop) didn't resurrect and hence rule 10-1074 binder out. Otherwise, scFv-Fc.r1 of 10-1074 has potency even better than the IgG version.

Response: Thank you for the comment. We have revised the statement to include the reduction in expression levels.

Revised text: “Designs for 10-1074 still showed a ~6-fold reduction in expression levels compared to parental IgG (Fig. 2A, C) and were no longer pursued due to the success with the PGT121 single-chain designs (Fig, 2C).”

  • Line 255-256: As compared to the IgG or scFv-Fc.r1? Authors must be comparing it with scFv-Fc.r1, but for reader's benefit, it is better to mention it explicitly here as well.

Response: Thank you for this suggestion. We have updated the sentence for clarity.

Revised text:For PGDM1400, although there were several round 2 scFv-Fc point mutations with favorable expression and potency, we selected the wild-type VH-VL, with a 20-amino acid linker as the top candidate (scFv-Fc.r2; Fig. 2A), as expression of PGDM1400 increased from 200 µg/mL for the scFv-Fc.r1 variant to 340 µg/mL for the scFv-Fc.r2 variant, with potency remaining comparable to the parental IgG (Fig. 2B).”

  • Line 260: It is strongly suggested to add a table mentioning retention time (in min) and % monomer content of the desired peak for each of the IgGs and scFv-Fcs tested (both r1 and r2variants)

Response: Thank you for the comment. The retention time and % monomer content for the round 2 scFv-Fc variants has now been added to the SEC profiles shown in Figure 2.

  • Line 268-270: Restructuring of the sentence required.

Response: As suggested, this sentence has been restructured for clarity.

Revised text: “Round 2 designs containing additional mutations for all antibodies were engineered to improve protein thermostability and obtain in vitro expression levels comparable to their unmutated scFv-Fc counterparts”

  • Line 271-273: Authors should provide the information about what mutations were made (in the final chosen r2 variants) and the criteria of their selection. A table is highly recommended.

Response: A supplementary table (Table S2) has been included providing information on the mutations made in the scFV-Fc round 2 variants. This table has been referred to in line 281 of the revised manuscript. As mentioned in lines 280-282 of the revised manuscript, these mutations were selected to improve protein thermostability of the round 2 variants and obtain in vitro expression levels comparable to their unmutated scFv-Fc counterparts.

  • Line 276: Correct font size.

Response: The font size has been updated to match the rest of the manuscript

  • Line 280/281: Are the authors concerned about the rate of decrease in serum concentration of PGT121 scFv-Fc.r2 variant atd5 post-injection? Can hepatic accumulation be a reason for this decrease as systemically delivered mRNA–LNP complexes mainly target the liver? Can authors comment on this? Have the authors tried (or are they going to try) intradermal/intramuscular/subcutaneous administration of mRNA-LNPs as administration via these routes have shown to produce prolonged protein expression?

Response: In wild type mice, human antibodies can be cleared quite rapidly. In addition, thermostability of a protein can impact in vivo half-lives. These observed rates are expected and are consistent with other antibodies and platforms. We have previously completed a Phase 1 study using mRNA encoded anti-chikungunya antibody (August A et al. Nat Med. 2021 Dec;27(12):2224-2233) where the half-life of a LS mutant human IgG was shown to be several months, suggesting no antibody sink in the liver. We have also tested other routes of administration; however, these data are unpublished. 

  • Line 282: Was dosing of N6 IgG mRNA-LNP the same (0.5mg/kg)?

Response: Yes, the round 2 designs for PGT121 and PGDM1400, as well as the full-length N6 IgG were administered at a concentration of 0.5 mg/Kg. The text has been updated to include the dosing of the N6 IgG mRNA-LNP.

Revised text: Full length N6 IgG (0.5 mg/Kg) achieved peak antibody titers…”

  • Line 284: How many variants of each were taken into in vivo expression?

Response: In vitro, we tested more than 20 variants for each bnAb for expression and neutralization; however, only variants that met certain criteria were tested in vivo (5-10 per bnAb). Data are shown for the variants with the most reproducible or robust dataset.

  • Line 292-295: Have the authors measured the self-aggregation propensity of the chosen bnAbs (for example by AC-SINS)?

Response: Thank you for the comment. Self-aggregation propensity was not measured as this is typically only relevant at high protein concentrations (>5mg/mL). We hypothesize that mRNA administration would lead to slow accumulation of protein over time in animals and would likely overcome aggregation induced by high protein concentrations.

  • Line 307: Just out of curiocity, why did the authors use hemizygous Tg32 mice instead of homozygous Tg32 (hFcRn)? Overall, the data presented by the authors adds value to the mRNA-LNP landscape and successfully shows the in vivo delivery of mRNA-LNPs and production of multiple antibodies (IgGs andscFv-Fcs). The anti-HIV1 antibodies targeting different epitopes ofHIV1 successfully neutralized the pseudovirus in vitro. The authors however did not test the protection offered by such passive immunization by challenging (mRNA-LNP)-injected mice with HIV virus (to demonstrate the in vivo efficacy offered by such injections). That would have been really convincing

Response: At the time of this study, Tg32 hemizygous mice were considered appropriate for testing PK of human antibodies in mice, with a strong correlation to non-human primates and humans as described in Proetzel G et al. Methods. 2014 Jan 1; 65(1): 148–153